# In Silico Screening of Prospective MHC Class I and II Restricted T-Cell Based Epitopes of the Spike Protein of SARS-CoV-2 for Designing of a Peptide Vaccine for COVID-19

**Kishore Sarma [1], Nargis K. Bali [2], Neelanjana Sarmah [3] and Biswajyoti Borkakoty [3,*]**

[1] Department of Zoology, Guwahati University, Assam 781014, India
[2] Department of Clinical Microbiology, Sher-i-Kashmir Institute of Medical Sciences, Soura, Srinagar, Jammu & Kashmir 190011, India
[3] Regional Viral Research and Diagnostic Laboratory, ICMR-Regional Medical Research Centre (ICMR-RMRC), North East Region, Dibrugarh, Assam 786010, India
[*] Correspondence: biswaborkakoty@gmail.com; Tel.: +91-943-513-1316

**Abstract:** Multiple vaccines were developed and administered to immunize people worldwide against SARS-CoV-2 infection. However, changes in platelet count following the course of vaccination have been reported by many studies, suggesting vaccine-induced thrombocytopenia. In this context, designing an effective targeted subunit vaccine with high specificity and efficiency for people with low platelet counts has become a challenge for researchers. Using the in silico-based approaches and methods, the present study explored the antigenic epitopes of the spike protein of SARS-CoV-2 involved in initial binding of the virus with the angiotensin converting enzyme-2 receptor (ACE-2) on the respiratory epithelial cells. The top ten major histocompatibility complex-I (MHC-I) and MHC-II restricted epitopes were found to have 95.26% and 99.99% HLA-class-I population coverage, respectively. Among the top ten promiscuous MHC-I restricted epitopes, 'FTISVTTEI' had the highest global HLA population coverage of 53.24%, with an antigenic score of 0.85 and a docking score of −162.4 Kcal/mol. The epitope 'KLNDLCFTNV' had the best antigenic score of 2.69 and an HLA population coverage of 43.4% globally. The study predicted and documented the most suitable epitopes with the widest global HLA coverage for synthesis of an efficient peptide-based vaccine against the deadly COVID-19.

**Keywords:** HLA; MHC; SARS-CoV-2; spike protein; T-cell epitopes; B-cell epitopes; vaccine; thrombocytopenia; platelet

## 1. Introduction

The SARS-CoV-2 genome comprises of four major structural proteins, namely, the spike (S), membrane (M), envelope (E), and nucleocapsid (N) proteins, together with a few accessory proteins and the non-structural proteins (NS1-16) coded by the ORFs [1–3]. The densely glycosylated spike (S) protein helps with its entry into host cells after interacting with the human angiotensin-converting enzyme 2 (ACE-2) receptor. The binding affinity of the ACE-2 receptor with the SARS-CoV-2 S ectodomain is much higher (approximately ten-to-twenty-fold higher affinity) in comparison to the S ectodomain of SARS-CoV [4–6]. Notably, the spike (S) protein of the SARS-CoV-2 contains some neutralizing epitopes crucial for the generation of antipathogenic (viral) immune responses of B cells [2,7]. As such, it is being used as an antigen for vaccine design by targeting at virus binding/fusion-blocking antibodies for neutralization of the viral infection [8]. Evidence suggests that multiple mutations responsible for altering the antigenic phenotype of SARS-CoV-2 and its variants and sub-variants are circulating across the world and thereby affecting the immune recognition by the host [9]. Human Leukocyte antigen (HLA) genes or Major histocompatibility complex (MHC class I and class II) are responsible for regulating the

responses of the host immune system towards the attack of viral pathogens because of their capacity to process and present antigenic peptides to T cells [10]. While class I HLA molecules, i.e., A, B, C are recognized by cytotoxic CD8+ T cells, the class II HLA molecules (HLA-DR, DQ, and DP) bind to extracellular viral antigen peptides thereby presenting them to CD4+ cells. This leads to the production of specific antibodies through the stimulation of B cells [11]. Extraordinary polymorphisms and different alleles of HLA genes deeply influence the property of HLA molecules to bind to specific peptides, which eventually contributes towards the specific responses of the host immune system against a pathogen or a vaccine [12]. Notably, peptides are emerging as crucial vaccine candidates due to their adequate stability, easy synthesis, and inability to cause a specific infection [13]. For a better understanding of vaccine efficacy and the implications of antigenic variations it is essential to determine the vaccine status and virus sequences [9].

Believing that specifically designed vaccines might play an essential role in the protection of an individual life-threatening infection of SARS-CoV-2, many companies have come up with multiple vaccines, resulting in the administration of vaccines to billions of people across the world. However, considering the many adverse thrombotic effects reported post-vaccination, it has become pertinent to design a targeted vaccine that may be efficiently used against the multiple variants of SARS-CoV-2 without any major complications in people with a compromised immune system. Given that the present in silico study based on an immunoinformatic approach was performed to examine the HLA recognition of antigenic epitopes of the spike protein of COVID-19, the study predicted and documented the most suitable epitopes with the widest global HLA coverage for synthesis of an efficient peptide-based vaccine against the deadly COVID-19.

## 2. Materials and Methods

### 2.1. Screening of MHC Class-I and Class-II Epitopes

The reference amino acid sequence of the surface or spike glycoprotein of Wuhan-Hu-1 isolate of SARS-CoV-2 was taken from NCBI (accession no: YP_009724390). Immune Epitope Database (IEDB) (https://www.iedb.org/ (accessed on 30 July 2022) was used to identify the T-cell epitopes for coronavirus restricted to both MHC-I and II. A set of HLA class I alleles proposed by Weiskopf and co-workers [14] and a set of HLA class II alleles proposed by Greenbaum and co-workers [15] covering most of the global population was selected for T-cell epitope prediction. The spike glycoprotein sequence of the novel coronavirus was run and analyzed against the reference alleles using the algorithm as recommended by IEDB. The epitopes thus identified were chosen considering the percentile rank threshold. A threshold of less than or equal to 1 was chosen for class I MHC and one less than or equal to 20 was set for MHC class II.

### 2.2. Prediction of Suitable Epitopes on the Basis of Antigenicity

From the preliminary screening, the epitopes which possess sufficient binding capacity to bind with multiple HLA alleles within the set threshold were combined to exclude the duplicates. This non-redundant putative epitope dataset was screened for antigenicity using VaxiJenv2.0 server [16]. The threshold for probable antigenicity was set to the default value of 0.4. The population coverage calculation tool from IEDB was selected for computing the population coverage of probable antigenic epitopes on the basis of the frequency of (allelic frequency) of interacting alleles of the HLA molecule. Area(s) and/or population(s) were set to the world to calculate the population coverage. Through analysis of the population coverage, the 10 most promiscuous epitopes with the highest global population coverage were selected for further analysis.

HpepDock web server was used to dock the top ten MHC-I and II restricted promiscuous epitopes against the reference allele HLA-A*02:01 and HLA-DPA1*01:03, respectively, as most of the candidate epitopes showed their binding potency towards these alleles [17]. This server uses the hierarchical flexible docking approach. The web server uses MODPEP to generate ensemble conformations of the peptide. The reference alleles were downloaded

from PDB (PDB Id: 1I4F, class I MHC molecule and PDB Id: 3LQZ, class II MHC molecule) and prior to docking, missing hydrogen atoms were inserted and molecules of water, ions along with ligands thus removed from the PDB file. A tumor-specific 10 mer antigenic peptide was attached to the reference allele HLA-A*02:01 and a self-peptide of 15 mer with nine binding core positions (FHYLPFLPS) and six flanking residues (two towards the N terminus and four at the C terminus) were attached to the reference allele HLA-DPA1*01:03. The conformation of the peptides was used as a binding site for the docking process. Re-docking of the peptides was performed into the peptide binding groove to check the reliability of the docking software. A graphical representation of the methodology employed in the present study is shown in Figure 1. Separately, the predicted epitopes in the RBD region of spike protein were analyzed for antigenicity, HLA restrictions and affinity of binding for the class I and class II HLA allele, as described above.

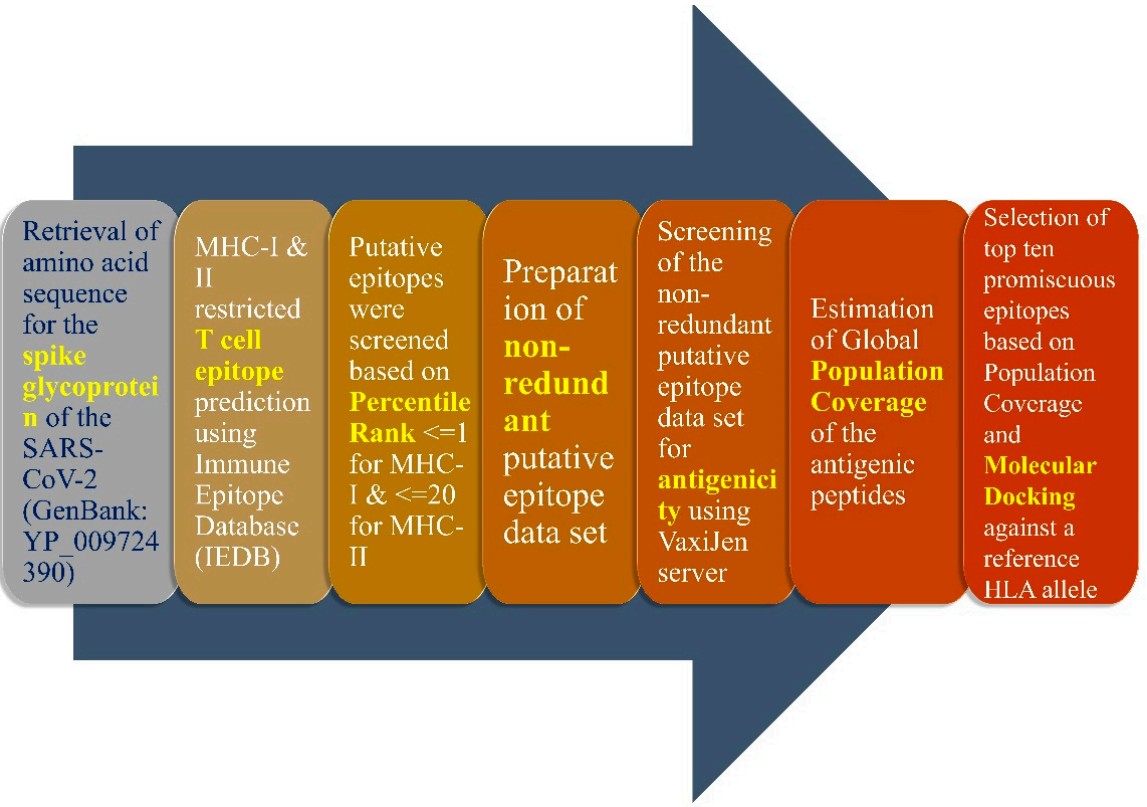

**Figure 1.** Graphical representation of the methodology used in the present study. The above-mentioned protocol was employed for identification of MHC-I and II restricted T-cell epitope, particularly those belonging to the receptor-binding domain (RBD) of the spike glycoprotein. The RBD of SARS-CoV-2 lies within the amino acid residues 331 to 524.

*2.3. B Cell-Based Epitope Prediction*

For the determination of the linear epitopes of B cells present in the spike protein (RBD domain) of COVID-19, the BepiPred 2.0 server was used. The threshold of 0.55, which corresponds to more than 0.817 specificity and less than 0.292 sensitivity, was chosen for the analysis. VaxiJen v2.0 server with 0.4 as the default threshold value was used to predict the antigenicity. DiscoTope 2.0 server was used to predict the discontinuous B-cell epitopes of RBD of SARS-CoV-2 spike protein using the three-dimensional structure of the SARS-CoV-2 Spike RBD [PDB ID: 6M0J, E chain] with a default threshold of −3.7, which corresponds to more than 0.75 specificity and less than 0.47 sensitivity. This was used to evaluate the antigenicity of the peptides that had more than eight residues.

## 3. Results

### 3.1. Selection of Class I and Class II MHC Restricted Epitopes Based on Population Coverage and Antigenicity

A total of 136566 MHC-I restricted prospective epitopes were identified by IEDB for the spike protein and among them, 647 were screened based on their percentile rank ($\leq$1). Further, 345 non-redundant epitopes were screened and scanned for their antigenicity, out of which 190 epitopes were found to be antigenic. Among the 190 antigenic epitopes, the top 10 promiscuous MHC-I restricted epitopes have been screened by checking the population coverage (Table 1). A single peptide 'FTISVTTEI' (HLA-B*58:01, HLA-A*68:02, HLA-A*02:06, HLA-B*51:01, HLA-A*26:01, HLA-A*02:03, HLA-A*02:01) showed a global HLA population coverage of 53.24% with an antigenic score of 0.85 and a docking score of $-162.4$ Kcal/mol, as presented in Table 1. Among the top 10 peptides based on population coverage, the epitope 'KLNDLCFTNV' has the best antigenic score of 2.69, docking score of $-166.5$ Kcal/mol and HLA population coverage of 43.4% globally. Meanwhile, the peptide 'YIWLGFIAGL' had the best docking score of $-182.4$ Kcal/mol, antigenicity of 0.58 and HLA population coverage of 40.6% globally. All 10 epitopes had a global population coverage of more than 39 % and their cumulative coverage was 95.26%. None of the epitopes showed perfect identity with the human proteome while performing BLAST against it.

**Table 1.** Top 10 promiscuous MHC-I restricted T-cell epitopes of spike glycoprotein of SARS-CoV-2 along with population coverage, cumulative population coverage, VaxiJen score and docking score against the reference allele HLA-A*02:01.

| Peptide | Allele | Population Coverage | Cumulative Population Coverage | VaxiJen Score | Docking Score Kcal/mol |
|---|---|---|---|---|---|
| **P1: FTISVTTEI** | HLA-A*68:02, HLA-B*58:01, HLA-A*02:06, HLA-A*26:01, HLA-B*51:01, HLA-A*02:01, HLA-A*02:03 | 53.24% | | 0.8535 | $-162.402$ |
| **P2: FAMQMAYRF** | HLA-B*35:01, HLA-B*53:01, HLA-A*23:01, HLA-B*58:01, HLA-A*24:02, HLA-B*08:01 | 47.12% | | 1.0278 | $-173.416$ |
| **P3: YQPYRVVVLSF** | HLA-A*23:01, HLA-A*24:02, HLA-B*07:02, HLA-B*51:01, HLA-B*15:01 | 46.30% | 95.26% | 0.8648 | $-178.359$ |
| **P4: SLIDLQELGK** | HLA-A*11:01, HLA-A*03:01, HLA-A*01:01 | 45.19% | | 1.0275 | $-131.419$ |
| **P5: KLNDLCFTNV** | HLA-A*02:03, HLA-A*32:01, HLA-A*02:01 | 43.40% | | 2.6927 | $-166.504$ |
| **P6: FVFLVLLPLV** | HLA-A*02:06, HLA-A*02:01, HLA-A*02:03, HLA-A*02:06, HLA-A*02:01, HLA-A*68:02 | 43.26% | | 0.8044 | $-170.516$ |

**Table 1.** *Cont.*

| Peptide | Allele | Population Coverage | Cumulative Population Coverage | VaxiJen Score | Docking Score Kcal/mol |
|---|---|---|---|---|---|
| **P7: KIADYNYKL** | HLA-A*32:01, HLA-A*02:01 | 42.66% | | 1.6639 | −164.515 |
| **P8: YIWLGFIAGL** | HLA-A*02:01, HLA-A*02:06 | 40.60% | | 0.5798 | −182.395 |
| **P9: GLIAIVMVTI** | HLA-A*02:03, HLA-A*02:01 | 39.84% | | 1.0813 | −159.635 |
| **P10: FELLHAPATV** | HLA-A*02:01 | 39.08% | | 0.5982 | −167.465 |

For MHC-II restricted prospective epitopes, a total of 33,993 epitopes of spike protein were identified by IEDB, and among them, 6320 were screened based on their percentile rank (≤20). Further, 385 non-redundant epitopes were selected based on their core peptide and among them, 190 were found to be antigenic. Among these antigenic peptides, the top 10 promiscuous MHC-II-restricted epitopes were screened by observing their population coverage, as shown in Table 2. All the epitopes had a global population coverage of more than 99%. Among the top 10 peptides, the epitope 'IRASANLAA' has the highest HLA-class-II population coverage of 99.94%, but has a low antigenic score of 0.44 and a moderately good docking score of −160.2 Kcal/mol. The epitope 'FLHVTYVPA' had the highest antigenic score, a high HLA-class-II population coverage as well as a docking score of 1.33, 99.9% and 204.5 Kcal/mol, respectively. Meanwhile, the epitope 'FSNVTWFHA' had the best docking score of −238.8 Kcal/mol, antigenic score of 0.81 and population coverage of 99.49%. None of the top 10 epitopes showed perfect identity with the human proteome while performing BLAST against it.

**Table 2.** Top 10 promiscuous MHC-II restricted T-cell epitopes of spike glycoprotein of SARS-CoV-2 along with population coverage, cumulative population coverage, VaxiJen score and docking score against the reference allele HLA-DPA1*01:03.

| Peptide | Allele | Population Coverage | Cumulative Population Coverage | VaxiJen Score | Docking Score Kcal/mol |
|---|---|---|---|---|---|
| **P1: IRASANLAA** | HLA-DPA1*02:01,HLA-DPB1*14:01,HLA-DQA1*01:02,HLA-DQB1*06:02,HLA-DRB1*09:01,HLA-DRB1*04:05,HLA-DPA1*01:03,HLA-DPB1*04:01,HLA-DRB1*11:01,HLA-DRB1*04:01,HLA-DRB1*01:01,HLA-DRB1*08:02,HLA-DPA1*03:01,HLA-DPB1*04:02,HLA-DQA1*01:01,HLA-DQB1*05:01,HLA-DRB1*12:01,HLA-DRB1*13:02,HLA-DRB1*07:01,HLA-DQA1*05:01,HLA-DQB1*03:01,HLA-DRB1*03:01,HLA-DQA1*04:01,HLA-DQB1*04:02,HLA-DQA1*03:01,HLA-DQB1*03:02 | 99.94% | 99.99% | 0.4455 | −160.809 |
| **P2: FLHVTYVPA** | HLA-DPA1*01:03,HLA-DPA1*02:01,HLA-DPA1*03:01,HLA-DPB1*01:01,HLA-DPB1*02:01,HLA-DPB1*04:01,HLA-DPB1*04:02,HLA-DPB1*05:01,HLA-DPB1*14:01,HLA-DQA1*01:01,HLA-DQA1*01:02,HLA-DQA1*03:01,HLA-DQA1*04:01,HLA-DQA1*05:01,HLA-DQB1*02:01,HLA-DQB1*03:02,HLA-DQB1*04:02,HLA-DQB1*05:01,HLA-DQB1*06:02,HLA-DRB1*04:01 | 99.90% | | 1.3346 | −204.550 |

**Table 2.** *Cont.*

| Peptide | Allele | Population Coverage | Cumulative Population Coverage | VaxiJen Score | Docking Score Kcal/mol |
|---|---|---|---|---|---|
| **P3: FNATRFASV** | HLA-DRB1*01:01, HLA-DPA1*01:03, HLA-DPB1*04:01, HLA-DPA1*02:01, HLA-DPB1*14:01, HLA-DRB1*15:01, HLA-DPB1*05:01, HLA-DRB1*04:01, HLA-DPA1*03:01, HLA-DPB1*04:02, HLA-DQA1*05:01, HLA-DQB1*03:01, HLA-DQA1*03:01, HLA-DQB1*03:02, HLA-DQA1*01:02, HLA-DQB1*06:02, HLA-DRB1*03:01, HLA-DQB1*02:01 | 99.85% | | 0.5609 | −199.001 |
| **P4: FTISVTTEI** | HLA-DPA1*01:03,HLA-DPA1*02:01,HLA-DPA1*03:01,HLA-DPB1*01:01,HLA-DPB1*04:01,HLA-DPB1*04:02,HLA-DQA1*01:02,HLA-DQA1*03:01,HLA-DQA1*05:01,HLA-DQB1*03:01,HLA-DQB1*03:02,HLA-DQB1*06:02,HLA-DRB1*01:01,HLA-DRB1*03:01,HLA-DRB1*04:01,HLA-DRB1*04:05,HLA-DRB1*07:01,HLA-DRB1*08:02,HLA-DRB1*11:01,HLA-DRB3*02:02 | 99.76% | | 0.8535 | −140.288 |
| **P5: FLPFFSNVT** | HLA-DPA1*01:03, HLA-DPA1*02:01, HLA-DPA1*03:01, HLA-DPB1*01:01, HLA-DPB1*02:01, HLA-DPB1*04:01, HLA-DPB1*04:02, HLA-DPB1*05:01, HLA-DPB1*14:01, HLA-DQA1*03:01, HLA-DQA1*05:01, HLA-DQB1*02:01, HLA-DQB1*03:01, HLA-DQB1*03:02, HLA-DRB1*01:01 | 99.68% | | 0.4400 | −183.885 |
| **P6: FSNVTWFHA** | HLA-DPA1*01:03, HLA-DPA1*02:01, HLA-DPA1*03:01, HLA-DPB1*01:01, HLA-DPB1*04:01, HLA-DPB1*04:02, HLA-DPB1*05:01, HLA-DPB1*14:01, HLA-DQA1*01:01, HLA-DQA1*04:01, HLA-DQA1*05:01, HLA-DQB1*03:01, HLA-DQB1*04:02, HLA-DQB1*05:01, HLA-DRB1*01:01, HLA-DRB1*03:01, HLA-DRB1*04:05 | 99.49% | | 0.8156 | −238.837 |
| **P7: FGAISSVLN** | HLA-DPA1*01:03, HLA-DPA1*02:01, HLA-DPA1*03:01, HLA-DPB1*02:01, HLA-DPB1*04:01, HLA-DPB1*04:02, HLA-DPB1*05:01, HLA-DPB1*14:01, HLA-DQA1*01:01, HLA-DQA1*01:02, HLA-DQA1*03:01, HLA-DQB1*03:02, HLA-DQB1*05:01, HLA-DQB1*06:02, HLA-DRB1*01:01 | 99.47% | | 0.5435 | −164.450 |
| **P8: FGAGAALQI** | HLA-DPA1*01:03, HLA-DPA1*02:01, HLA-DPA1*03:01, HLA-DPB1*01:01, HLA-DPB1*02:01, HLA-DPB1*04:01, HLA-DPB1*04:02, HLA-DPB1*05:01, HLA-DPB1*14:01, HLA-DQA1*01:02, HLA-DQA1*05:01, HLA-DQB1*03:01, HLA-DQB1*06:02, HLA-DRB1*09:01, HLA-DRB3*01:01 | 99.45% | | 0.6377 | −163.317 |
| **P9: FKIYSKHTP** | HLA-DPA1*01:03, HLA-DPA1*02:01, HLA-DPA1*03:01, HLA-DPB1*01:01, HLA-DPB1*02:01, HLA-DPB1*04:02, HLA-DPB1*14:01, HLA-DQA1*01:01, HLA-DQA1*03:01, HLA-DQA1*05:01,HLA-DQB1*03:01,HLA-DQB1*03:02,HLA-DQB1*05:01,HLA-DRB1*01:01,HLA-DRB1*03:01 | 99.44% | | 0.9886 | −171.594 |

**Table 2.** *Cont.*

| Peptide | Allele | Population Coverage | Cumulative Population Coverage | VaxiJen Score | Docking Score Kcal/mol |
|---------|--------|--------------------|-----------------------------|---------------|------------------------|
| **P10: FRVQPTESI** | HLA-DPA1*02:01,HLA-DPA1*03:01,HLA-DPB1*01:01,HLA-DPB1*04:02,HLA-DPB1*05:01,HLA-DPB1*14:01,HLA-DQA1*01:01,HLA-DQA1*01:02,HLA-DQA1*03:01,HLA-DQA1*04:01,HLA-DQA1*05:01,HLA-DQB1*03:01,HLA-DQB1*03:02,HLA-DQB1*04:02,HLA-DQB1*05:01,HLA-DQB1*06:02,HLA-DRB1*01:01,HLA-DRB1*03:01,HLA-DRB1*04:01,HLA-DRB1*04:05 | 99.19% | | 0.9396 | −155.782 |

### 3.2. Re-Docking of the Peptides Confirmed the Specificity of the Selected Peptides

Re-docking of the tumor-specific 10-mer antigenic peptide and 9-mer core peptide of the 15-mer self-peptide inside the peptide binding groove of MHC-I and II restricted reference allele showed an RMSD of 0.72 and 0.67 Å with the co-crystal conformation and conservancy within interactions and interacting residues (Figures 2 and 3). This strongly supports the docking protocol used for re-docking of the control peptides. Binding affinity analysis of the control peptides showed a binding affinity of −167.96 and −169.236 kcal/mol for MHC-I and II restricted alleles, respectively. Subsequent docking of the top 10 epitopes showed their binding affinity within the binding groove of peptides of both the reference alleles, as in the case of the control peptides (Figures 2 and 3). The binding affinity of the top 10 MHC-I and II restricted epitopes of T cells are presented in Tables 1 and 2, respectively.

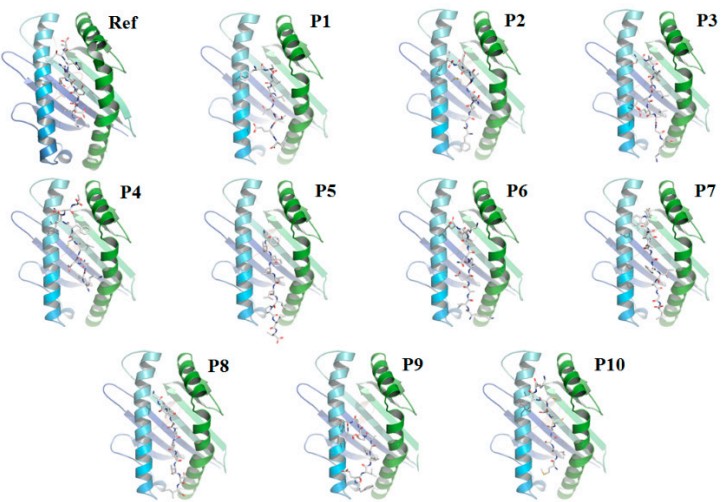

**Figure 2.** Docking of the tumor-specific 10-mer antigenic peptide (as control) and top 10 MHC-I restricted epitopes against HLA-A*02:01 allele (**Ref**) tumor-specific 10- mer antigenic peptide and HLA-A*02:01 (1) **P1**: FTISVTTEI and HLA-A*02:01 (2) **P2**: FAMQMAYRF and HLA-A*02:01 (3) **P3**: SLIDLQELGK and HLA-A*02:01 (4) **P4**: KLNDLCFTNV and HLA-A*02:01 (5) **P5**: FVFLVLLPLV and HLA-A*02:01 (6) **P6**: KIADYNYKL and HLA-A*02:01 (7) **P7**: YIWLGFIAGL and HLA-A*02:01 (8) **P8**: GLIAIVMVTI and HLA-A*02:01 (9) **P9**: FELLHAPATV and HLA-A*02:01 (10) **P10**: MTSCCSCLK and HLA-A*02:01.

### 3.3. T Cell Epitope Prediction with Restriction to Class I and Class II MHC Molecules

Subsequent determination of MHC-I restricted T-cell epitopes belonging to RBD suggested 20,035 putative epitopes, of which 129 were screened based on the percentile rank threshold for further analysis. Among these 129 putative epitopes, 62 non-redundant epitopes were screened for their antigenicity and 30 were found to be antigenic. Of

these 30 antigenic epitopes, the top 10 epitopes were screened on the basis of their global population coverage, which had cumulative population coverage of 94.54%. Table 3 shows the top 10 promiscuous MHC-I restricted epitopes of T cells from the SARS-CoV-2 RBD domain, along with population coverage, cumulative population coverage and VaxiJen score. Further, MHC-I restricted HLA-A*02:01 (Population coverage: 39.08%), HLA-A*24:02 (Population coverage: 21.38%) and HLA-A*23:01 (Population coverage: 5.43%) recognizes three epitopes each out of the top 10 epitopes of the RBD region (as shown in Table 3). The MHC-I restricted epitope 'KLNDLCFTNV' in the RBD region had the highest antigenic score of 2.69 among the top 10 epitopes analyzed, while the epitope 'YQPYRVVVLSF' had the highest HLA class I population coverage of 46%.

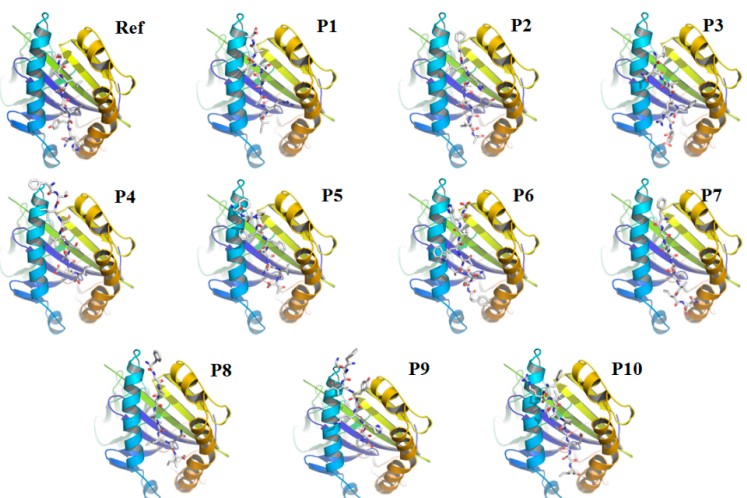

**Figure 3.** Docking of the core 9-mer of the self−15-mer peptide (as control) and top 10 MHC-II restricted epitopes against HLA-DPA1*01:03 allele. (**Ref**) 9-mer of the self-15-mer peptide and HLA-DPA1*01:03 (1) **P1:** IRASANLAA and HLA-DPA1*01:03 (2) **P2:** FLHVTYVPA and HLA-DPA1*01:03 (3) **P3:** FNATRFASV and HLA-DPA1*01:03 (4) **P4:** FTISVTTEI and HLA-DPA1*01:03 (5) **P5:** FLPFFSNVT and HLA-DPA1*01:03 (6) **P6:** FSNVTWFHA and HLA-DPA1*01:03 (7) **P7:** FGAISSVLN and HLA-DPA1*01:03 (8) **P8:** FGAGAALQI and HLA-DPA1*01:03 (9) **P9:** FKIYSKHTP and HLA-DPA1*01:03 (10) **P10:** FRVQPTESI and HLA-DPA1*01:03.

**Table 3.** Top 10 promiscuous MHC-I restricted T-cell epitopes belonging to the RBD of SARS-CoV-2 along with population coverage, cumulative population coverage and VaxiJen score.

| Peptide | Start-End | Allele | Population Coverage | Cumulative Population Coverage | VaxiJen Score |
|---|---|---|---|---|---|
| **P1: YQPYRVVVLSF** | 505–515 | HLA-A*23:01, HLA-A*24:02, HLA-B*07:02, HLA-B*51:01, HLA-B*15:01 | 46.30% | | 0.8648 |
| **P2: KLNDLCFTNV** | 386–395 | HLA-A*32:01, HLA-A*02:03, HLA-A*02:01 | 43.40% | | 2.6927 |
| **P3: KIADYNYKL** | 417–425 | HLA-A*32:01, HLA-A*02:01 | 42.66% | | 1.6639 |
| **P4: FELLHAPATV** | 515–524 | HLA-A*02:01 | 39.08% | | 0.5982 |
| **P5: FASVYAWNRK** | 347–356 | HLA-A*68:01, HLA-A*33:01, HLA-A*11:01, HLA-A*31:01 | 27.17% | 94.54% | 0.5868 |
| **P6: CYFPLQSYGFQ** | 488–498 | HLA-A*24:02, HLA-A*23:01 | 26.18% | | 0.7378 |
| **P7: VYAWNRKRI** | 350–358 | HLA-A*24:02, HLA-A*23:01 | 26.18% | | 0.5003 |
| **P8: RQIAPGQTGK** | 408–417 | HLA-B*15:01, HLA-A*03:01 | 23.84% | | 1.7893 |
| **P9: QTGKIADYNY** | 414–423 | HLA-A*30:02, HLA-A*01:01 | 19.55% | | 1.5116 |
| **P10: NLDSKVGGNY** | 440–449 | HLA-A*01:01 | 17.34% | | 0.7882 |

Similarly, the identification of MHC-II restricted T cell epitopes belonging to RBD suggested 4860 putative epitopes out of which 931 were screened based on the percentile rank threshold for further analysis. Out of these 931 putative epitopes, 98 non-redundant epitopes were screened for their antigenicity and 46 were found to be antigenic. Among these 46 antigenic epitopes, the top 10 epitopes were selected based on their global population coverage, amounting to a cumulative population coverage of 99.99%. Table 4 shows the top 10 promiscuous class IMHC-restricted epitopes of T cells from the SARS-CoV-2 RBD domain. The table also shows the population coverage, cumulative population coverage and VaxiJen score of the epitopes. The MHC-II restricted epitope 'VVVLSFELLH' in the RBD region had the highest antigenic score of 1.22 and population coverage of 98.9% among the top 10 epitopes analyzed, while the epitope 'FNATRFASV' had the highest single-epitope HLA-II population coverage of 99.85%. Out of 10 MHC-II limited epitopes of T cells belonging to RBD of COVID-19, HLA-DPA1*01:03 (Population coverage: 69.75%) was targeted by all ten epitopes (Table 5).

### 3.4. Prediction and Determination of Linear and Discontinuous Epitopes of B Cells

For the prediction of linear epitopes of B-cell (RBD domain of spike protein of COVID-19), IEDB together with the BepiPred 2.0 server was used. The residues with scores above 0.5 were predicted to be part of an epitope. Of the epitopes detected through the server, the five-best linear (continuous) epitopes of B cell were selected based on a threshold of 0.55, which corresponds to more than 0.817 specificity and less than 0.292 sensitivity (as shown in Figure 4). The peptide length of all five epitopes ranged from 9–15, respectively, as presented in Table 6. These five predicted models were further assessed by the BepiPred 2.0 server, and the antigenicity of the predicted epitopes was examined on the basis of the scores obtained from VaxiJen v2.0 server, with 0.4 as the default threshold value. It is worth mentioning that among the top five peptides obtained from the server, the epitope 'TGKIADYNYKLP' has the best antigenic score of 1.19, followed by 'DEVRQIAPG' peptide with a VaxiJen v2.0 score of 0.72 (as shown in Table 6) indicating the potential of developing immunogenicity. However, the other three peptides showed the VaxiJen score ranging from 0.09–0.39, respectively.

**Table 4.** Top 10 promiscuous MHC-II restricted T-cell epitopes belonging to the RBD of SARS-CoV-2 along with population coverage, cumulative population coverage and VaxiJen score.

| Peptide | Start-End | Allele | Population Coverage | Cumulative Population Coverage | VaxiJen Score |
|---|---|---|---|---|---|
| **P1: FNATRFASV** | 342–350 | HLA-DRB1*01:01, HLA-DPA1*01:03, HLA-DPB1*04:01, HLA-DPA1*02:01, HLA-DPB1*14:01, HLA-DRB1*15:01, HLA-DPB1*05:01, HLA-DRB1*04:01, HLA-DPA1*03:01, HLA-DPB1*04:02, HLA-DQA1*05:01, HLA-DQB1*03:01, HLA-DQA1*03:01, HLA-DQB1*03:02, HLA-DQA1*01:02, HLA-DQB1*06:02, HLA-DRB1*03:01, HLA-DQB1*02:01 | 99.85% | | 0.5609 |
| | | | | 99.99% | |
| **P2: VVVLSFELLH** | 510–519 | HLA-DPA1*01:03, HLA-DPA1*02:01, HLA-DPA1*03:01, HLA-DPB1*01:01, HLA-DPB1*02:01, HLA-DPB1*04:02, HLA-DPB1*05:01, HLA-DQA1*01:01, HLA-DQA1*05:01, HLA-DQB1*02:01, HLA-DQB1*05:01, HLA-DRB1*03:01, HLA-DRB1*04:05, HLA-DRB1*09:01, HLA-DRB4*01:01 | 98.89% | | 1.2274 |

| Peptide | Start-End | Allele | Population Coverage | Cumulative Population Coverage | VaxiJen Score |
|---|---|---|---|---|---|
| **P3: RVVVLSFEL** | 509–517 | HLA-DPA1*01:03, HLA-DPA1*02:01, HLA-DPA1*03:01, HLA-DPB1*02:01, HLA-DPB1*04:01, HLA-DPB1*04:02, HLA-DPB1*05:01, HLA-DPB1*14:01, HLA-DRB1*07:01, HLA-DRB1*15:01 | 98.24% | | 1.1918 |
| **P4: YRVVVLSFE** | 508–516 | HLA-DPA1*01:03, HLA-DPA1*02:01, HLA-DPB1*04:01, HLA-DPB1*05:01, HLA-DPB1*14:01, HLA-DQA1*03:01, HLA-DQA1*05:01, HLA-DQB1*02:01, HLA-DQB1*03:02, HLA-DRB1*04:05, HLA-DRB4*01:01 | 98.06% | | 1.2096 |
| **P5: YQPYRVVVL** | 505–513 | HLA-DPA1*01:03, HLA-DPA1*02:01, HLA-DPA1*03:01, HLA-DPB1*01:01, HLA-DPB1*02:01, HLA-DPB1*04:01, HLA-DPB1*04:02, HLA-DRB1*01:01, HLA-DRB1*07:01, HLA-DRB1*15:01, HLA-DRB3*01:01 | 97.88% | | 0.5964 |
| **P6: FRKSNLKPF** | 456–464 | HLA-DPA1*01:03, HLA-DPA1*02:01, HLA-DPA1*03:01, HLA-DPB1*04:01, HLA-DPB1*04:02, HLA-DPB1*05:01, HLA-DPB1*14:01, HLA-DRB1*07:01, HLA-DRB1*08:02, HLA-DRB1*09:01, HLA-DRB1*11:01, HLA-DRB1*12:01, HLA-DRB1*13:02, HLA-DRB3*01:01, HLA-DRB3*02:02, HLA-DRB5*01:01 | 97.72% | | 0.6280 |
| **P7: VLYNSASFS** | 367–375 | HLA-DPA1*01:03, HLA-DPA1*02:01, HLA-DPB1*02:01, HLA-DPB1*14:01, HLA-DQA1*05:01, HLA-DQB1*03:01, HLA-DRB1*03:01, HLA-DRB1*08:02, HLA-DRB1*12:01, HLA-DRB1*13:02, HLA-DRB3*02:02 | 96.51% | | 0.4029 |
| **P8: VLSFELLHA** | 512–520 | HLA-DPA1*01:03, HLA-DPA1*02:01, HLA-DPB1*04:01, HLA-DPB1*14:01, HLA-DQA1*01:01, HLA-DQB1*05:01, HLA-DRB1*01:01, HLA-DRB1*04:01, HLA-DRB1*12:01, HLA-DRB1*15:01 | 96.07% | | 1.0776 |
| **P9: LCFTNVYAD** | 390–398 | HLA-DPA1*01:03, HLA-DPB1*04:01, HLA-DQA1*01:02, HLA-DQA1*03:01, HLA-DQA1*04:01, HLA-DQB1*03:02, HLA-DQB1*04:02, HLA-DQB1*06:02, HLA-DRB1*07:01 | 95.16% | | 0.9994 |
| **P10: FNCYFPLQS** | 486–494 | HLA-DPA1*01:03, HLA-DPB1*04:01, HLA-DQA1*01:01, HLA-DQB1*05:01, HLA-DRB1*01:01, HLA-DRB1*04:01, HLA-DRB1*04:05, HLA-DRB1*09:01, HLA-DRB1*15:01, HLA-DRB5*01:01 | 92.24% | | 1.0649 |

Discontinuous B-cell epitopes were determined using DiscoTope 2.0 server. As per the prediction obtained from the server, twenty-three discontinuous B-cell epitopes were observed with a default threshold of −3.7, as presented in Figure 5. The green peaks in the figure correspond to the positively predicted epitopes above the threshold line, and those of pink color slopes are predicted as the non-epitopic region of the protein. Of these

epitopes, ten residues—namely N439, K444, G502, G446, T500, Y449, Q498, N501, G496 and Y505—are known to be involved in the binding of human ACE2 receptor with RBD of the SARS-CoV-2.

**Table 5.** Frequency of HLA allele for MHC-II restricted T-cell epitopes (top 10) belonging to the RBD of SARS-CoV-2.

| HLA Allele | No of Hits | Epitopes | HLA Allele | No of Hits | Epitopes |
|---|---|---|---|---|---|
| HLA-DPA1*01:03 | 10 | P1-P10 | HLA-DPB1*02:01 | 4 | P2, P3, P5, P7 |
| HLA-DPB1*04:01 | 8 | P1, P3-P6, P8-P10 | HLA-DQA1*01:01 | 3 | P2, P8, P10 |
| HLA-DPA1*02:01 | 8 | P1-P8 | HLA-DQB1*02:01 | 3 | P1, P2, P4 |
| HLA-DPB1*14:01 | 6 | P1, P3, P4, P6-P8 | HLA-DRB1*04:05 | 3 | P2, P4, P10 |
| HLA-DRB1*15:01 | 5 | P1, P3, P5, P8, P10 | HLA-DRB1*04:01 | 3 | P1, P8, P10 |
| HLA-DPB1*05:01 | 5 | P1-P4, P6 | HLA-DRB1*12:01 | 3 | P6-P8 |
| HLA-DPA1*03:01 | 5 | P1-P3, P5, P6 | HLA-DRB1*09:01 | 3 | P2, P6, P10 |
| HLA-DPB1*04:02 | 5 | P1-P3, P5, P6 | HLA-DQA1*03:01 | 3 | P1, P4, P9 |
| HLA-DQA1*05:01 | 4 | P1, P2, P4, P7 | HLA-DQB1*03:02 | 3 | P1, P4, P9 |
| HLA-DRB1*01:01 | 4 | P1, P5, P8, P10 | HLA-DQB1*05:01 | 3 | P2, P8, P10 |
| HLA-DRB1*07:01 | 4 | P3, P5, P6, P9 | HLA-DRB1*03:01 | 3 | P1, P2, P7 |
| HLA-DRB5*01:01 | 2 | P6, P10 | HLA-DPB1*01:01 | 2 | P2, P5 |
| HLA-DRB3*01:01 | 2 | P5, P6 | HLA-DQB1*06:02 | 2 | P1, P9 |
| HLA-DRB1*08:02 | 2 | P6, P7 | HLA-DRB3*02:02 | 2 | P6, P7 |
| HLA-DQB1*03:01 | 2 | P1, P7 | HLA-DQA1*04:01 | 1 | P9 |
| HLA-DRB4*01:01 | 2 | P2, P4 | HLA-DRB1*11:01 | 1 | P6 |
| HLA-DQA1*01:02 | 2 | P1, P9 | HLA-DQB1*04:02 | 1 | P9 |
| HLA-DRB1*13:02 | 2 | P6, P7 | | | |

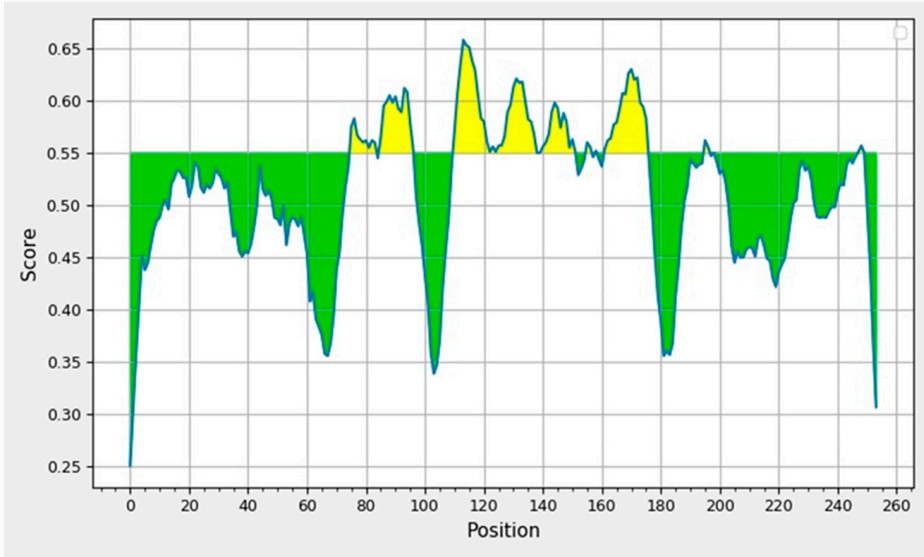

**Figure 4.** Displays the graphical representation of the epitopic and nonepitopic region of the B-cell epitopes of RBD domain of SARS-CoV-2 spike protein. The BepiPred-2.0 server predicts B-cell epitopes from a protein sequence, using a Random Forest algorithm trained on epitopes and non-epitope amino acids determined from crystal structures. The residues with scores above the threshold are predicted to be part of an epitope and colored in yellow on the graph where Y-axes depict amino acid residue scores and X-axes amino acid residue positions in the sequence.

**Table 6.** The best linear (continuous) B-cell epitopes of the RBD domain of SARS-CoV-2 spike protein in IEDB with a threshold of 0.55, which corresponds to more than 0.817 specificity and less than 0.292 sensitivity. BepiPred 2.0 was used to predict the linear B-cell epitopes of the RBD domain of SARS-CoV-2.

| Sl. No. | Start | End | Peptide | Peptide Length | VaxiJen Score (Cut Off = 0.4) |
|---|---|---|---|---|---|
| **1** | 405 | 413 | DEVRQIAPG | 9 | 0.7216 |
| **2** | 415 | 426 | TGKIADYNYKLP | 12 | 1.1956 |
| **3** | 440 | 468 | NLDSKVGGNYNYLYRLFRKSNLKPFERDI | 29 | 0.3934 |
| **4** | 470 | 481 | TEIYQAGSTPCN | 12 | 0.0966 |
| **5** | 491 | 505 | PLQSYGFQPTNGVGY | 15 | 0.3415 |

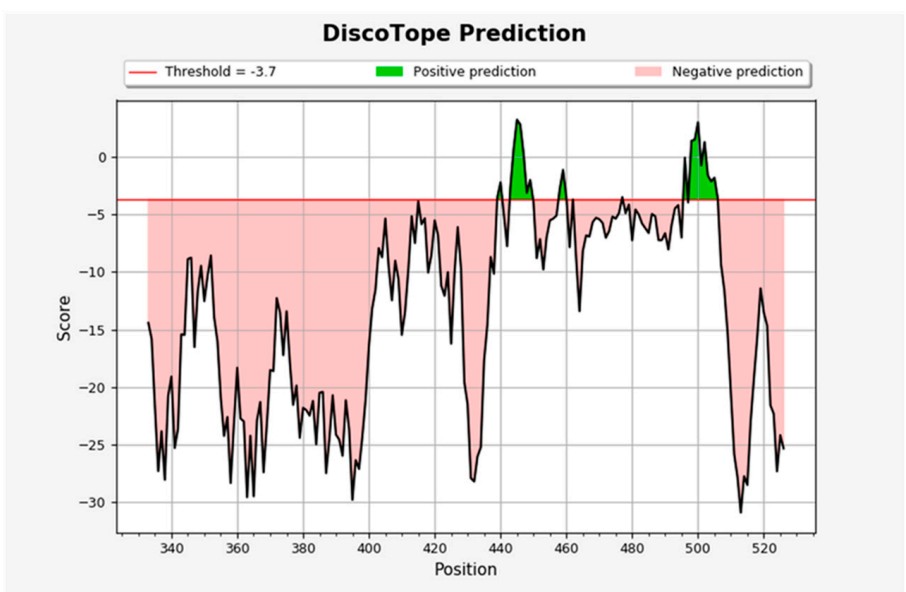

**Figure 5.** The discontinuous B-cell epitopes **N439**, N440, S443, **K444**, V445, **G446**, G447, N448, **Y449**, K458, S459, N460, S477, **G496**, **Q498**, P499, **T500**, **N501**, **G502**, V503, G504, **Y505**, Q506. Residues that are involved in binding of the SARS-CoV-2 RBD to hACE2 are shown in bold. The BepiPred-2.0 server predicts B-cell epitopes from a protein sequence, using a Random Forest algorithm trained on epitopes and non-epitope amino acids determined from crystal structures. The residues with scores above the threshold are predicted to be part of an epitope and colored in yellow on the graph where Y-axes depict amino acid residue scores and X-axes amino acid residue positions in the sequence.

## 4. Discussion

Synthesis of an efficient epitope-based peptide vaccine depends on the knowledge of immunodominant epitopes. The genes that encode the MHC molecules are highly polymorphic, varying substantially between individuals [18]. Different individuals are characterized by distinct HLA profiles and hence present different epitopes of the same pathogen to T-cells, thereby initiating a unique and different immune response [18]. As with many other infectious diseases, COVID-19 possesses a wide range of disease symptoms starting from asymptomatic cases to mild to severe infections and death [19]. Wide variation in the severity of diseases and mortality rates across different population is being noticed during the COVID-19 pandemic, in which co-morbid conditions such as cardiovascular diseases, hypertension, diabetes, etc., are considered to play a key role [20]. However, it is important to mention here that the role of HLA in individuals susceptible to severe infection, and death cannot be ruled out in the current scenario. As reported earlier, the spike protein of the coronavirus is also immunogenic and develops high IFN-γ-mediated

T-cell response [21], and SARS-CoV-2 is also capable of eliciting a high immunogenic T-cell response. It is also reported that immune responses generated via T cells towards COVID-19 spike peptides markedly enhanced after the vaccination process, leading to co-expression of INF-γ and TNF-α [22]. Our study used the Immune Epitope Database (IEDB), which is a repository for the data in relation to epitopes selected from the previously published literature, as well as for data generated from the epitopes discovery contracts of T and B cells funded by NIH in the year 2003 [18]. The present study showed that the top ten MHC-I restricted epitopes of spike protein had an HLA-class-I global population coverage of 95.2%, among which HLA-A*02:01 recognized seven epitopes out of the top ten MHC-I restricted epitopes (Table 1). The MHC-II restricted top ten epitopes had almost 100% global population coverage. Thus, peptide vaccines based on the epitopes of the spike protein should generate an immune response covering over 95% of the global population. Clearance of the virus during the preliminary course of viral infection is dependent on the effective immune response of T cells and its subtypes, i.e., CD4+ and CD8+ [23], and low peripheral T cell count is linked with non-survival of the patients with SARS-CoV-2 [24–26]. In our study, it was observed that at least 30 MHC-I restricted epitopes in RBD were antigenic, with the epitope 'KLNDLCFTNV' having the highest antigenic score. Additionally, the global population coverage was 94.5% for the top ten MHC-I restricted epitopes. HLA-DPA1*01:03, HLA-DPB1*04:01 and HLA-DPA1*02:01 are important MHC-II molecules recognizing multiple epitopes of the RBD region. Considering this, it is pertinent that populations with these prevalent HLA-II alleles should have a good immunogenic response against SARS-CoV-2 spike protein. Our finding was in accordance with a previously published study in which authors also documented that HLA-A* 0201 recognizes multiple MHC-I epitopes of spike protein, apart from other HLA Class-I alleles [27]. Our contention of targeting T-cell epitopes of spike protein of COVID-19 for the development of an effective vaccine was further supported by a recently published article in which the authors reported that the predicted CD8+ epitopes clustered in a region at ~550 amino acid of the spike glycoprotein, and also within two regions of ORF1ab polypeptide, respectively [28]. In addition, an immunoinformatic study by Almofti et al., showed that T-cell epitopes of spike protein 47-FRSDTLYLT-55, 195-YVYKGYQPI-203 and 880-FAMQMAYRF-888 of SARS-CoV (SARS) were documented to interact with both types of MHC alleles [29]. Of the three epitopes, our study also documented the epitope FAMQMAYRF of SARS-CoV-2, recognized by various MHC-I restricted HLA alleles including HLA-A* 0201. In an earlier study, Grifoni et al. analyzed and identified 1400 different epitopes based on T cells of COVID-19 and revealed distinct viral immunodominant regions and the prevalently recognized epitopic regions of SARS-CoV-2 [30]. A recently published study of phase I open-label trial indicated CoVac-1 as the efficient multi-peptide vaccine candidate responsible for inducing sufficient SARS-CoV-2 T-cell immunity, even in immunocompromised patients [31]. In addition, a recent retrospective cohort study revealed that BNT162b2 vaccine showed effectivity up to six months after being fully vaccinated, irrespective of widespread dissemination of the delta variant. However, the authors documented that the reduced efficacy of the vaccine against SARS-CoV-2 infection over time may be related to compromised immunity during the course of time, rather than escaping protection by the delta variant [32]. In a recent research article, a proprietary algorithm, OncoPeptVAC, was designed and developed by the authors for the determination of activating epitopes of CD8+ T cells. On testing of a cocktail comprised of eleven peptides (15-mer) that have wide coverage of class-I and class-II MHC genes, along with favorable TCR engagement predicted via the algorithm for activation of T cells in healthy donor individuals from the United States of America and India who were not exposed to the deadly virus, a higher CD8 T-cell activation was observed in comparison to the overlapping peptide pools (15-mer) of S1 and S2 spike proteins. This suggested the involvement of either one or more than one peptide in the pool showing cross-reactive TCR of other viruses not specifically from a coronavirus [33].

B-cell antigenicity is triggered by the antigen abundance of the targeted antigen during the course of infection, and the way it is being projected to the immune system [34]. Our

findings on B cell epitopes are in agreement with a recent article in which the authors also explained the prediction of linear epitopes of B cells in the spike protein of SARS-CoV-2 [35]. A previously published article reported that B cell epitopes that are linear in nature may be considered more suitable candidates to initiate a protective antibody response [36]. Contradictorily, a previous study showed that the antibody extracted from the COVID-19 positive individuals neutralized the virus by binding to the N-terminal domain and not to the RBD domain of the Spike protein [37]. Immunogenic epitopes of the Spike protein using linear peptide residues have been identified previously, and the antigenicity has also been validated using mice models [38,39]. Our data on the discontinuous epitope through the Disco Tope server revealed twenty-three predicted epitopes, including N501, K444, G446, S477 and N439 residues of the Spike protein of SARS-CoV-2. Here, it is worth mentioning that different mutations in positions such as N501, K444, G446, S477 and N439 are reported to enhance the affinity of these residues towards the binding of human ACE2 receptor [26,40]. In accordance with our study, a few recently published articles also documented the promising role of spike protein and T-cell-based epitopes in vaccine design for COVID-19 infection [41,42]. A recently published article by Zannella et al., 2022, showed that synthesized peptides from the selected nucleotide sequences of the genome of coronaviruses were not found to be toxic to the cultured cells and also had resistant properties against serum proteases. The study thus showed that due to the ability of these peptides to inhibit the infection of coronaviruses by binding to the RBD of the spike protein, these peptides may be used as promising targets for vaccine development [43]. It is essential to mention here that a previously published article used a comparative modeling approach, i.e., RosettaMHC, for prediction of the accurate SARS-CoV-2 CD8 epitope 3D models by analyzing the existing structures of peptide complexes present in the protein database [44]. In this article, the authors performed the modeling of epitopic peptides of 8–10 residues, which were predicted to bind to the common allele HLA-A*02:01. They also compared the electrostatic surfaces of models of homologous peptide/HLA-A*02:01 complexes from the coronavirus strains for determination and identification of suitable epitopes recognized by the cross-reactive T-cell receptors or TCRs [44]. A few studies have also predicted the epitopic regions of T cell epitopes for the determination of suitable vaccine candidates [45,46].

Platelets play a major role in the regulation of immune responses in COVID-19, and SARS-CoV-2 is known to interfere with the platelet functions leading to the development of thrombotic events. Given a similar scenario, vaccines are also reported to elicit thrombocytopenic effects inside the body after specific doses. It is suspected that the Spike protein (surface glycoprotein of SARS-CoV-2) secreted by muscle cells after mRNA vaccines or vector-based vaccines may result in the binding of free spike protein antigens in platelet surface receptors. This may lead to immune-mediated destruction (immune-complex formation) of platelets in persons generating spike-protein-specific antibodies following vaccination. A detailed review on vaccine development was presented in a previously published article in which the authors beautifully explained the methods of vaccine synthesis, as well as the benefits, strengths, associated risks and flaws of various types of vaccines developed and administered worldwide [47]. Although the detailed definite mechanism of COVID-19 or vaccine-mediated dysregulation of platelets has not been clearly identified, many studies have shown the morphological and functional changes of platelets in response to COVID-19 infection [48,49]. Vaccine-Induced Thrombotic Thrombocytopenia (VITT) is defined as an autoimmune condition of the development of antibodies to PF-4 after the administration of vaccine counterparts inside the body, which leads to activation of platelets leading to thrombosis in the arterial and venous circulation [50]. These observations are indicative of the fact that the spike protein of the virus is able to elicit the production of antiplatelet factor 4 antibodies, leading to ectopic activation of platelets. Thus, designing a peptide vaccine that evades the production of anti-PF4 antibodies is essential.

## 5. Conclusions

Our study on the prediction of epitopes (B and T cell) of the spike protein of SARS-CoV-2 provides a framework leading to the identification of suitable epitope-based vaccine candidates for better management of the disease. In the future, targeting specific T-cell-based epitopes may prove to be effective against the changing variants of SARS-CoV-2 and may also help with the development of sufficient T-cell immunity for immune-compromised patients.

**Author Contributions:** Conceptualization, B.B. and N.K.B.; methodology, B.B., N.K.B., K.S. and N.S.; software, K.S. and N.S.; validation, B.B., N.K.B., K.S. and N.S.; formal analysis, K.S. and N.S.; investigation, B.B., N.K.B., K.S. and N.S.; resources, B.B.; data curation, B.B. and K.S.; writing—original draft preparation, B.B., N.K.B. and K.S.; writing—review and editing, B.B, N.K.B., K.S. and N.S.; visualization, K.S.; supervision, B.B.; project administration, B.B.; funding acquisition, None. All authors have read and agreed to the published version of the manuscript.

**Funding:** This research received no external funding.

**Data Availability Statement:** Not applicable.

**Acknowledgments:** The authors would like to acknowledge the Regional Viral Research Diagnostic Laboratory (Regional VRDL), Dibrugarh, Department of Health Research (DHR), Govt. of India for providing the facility to carry out the present work.

**Conflicts of Interest:** The authors declare no conflict of interest.

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
