# Peer review of "In Silico Screening of Prospective MHC Class I and II Restricted T-Cell Based Epitopes of the Spike Protein of SARS-CoV-2 for Designing of a Peptide Vaccine for COVID-19"

_covid, doi:10.3390/covid2120124_

Round 1

Reviewer 1 Report

The manuscript describes in silico predicted antigenic epitopes of SARS-CoV-2, and the proportion of MHC I and II coverage of predicted peptides. The article is nicely written, however, there are some comments regarding Figures and spelling, which would be beneficial for the manuscript and readers to address before the paper is published.

Major comments.

1. Text in Figure 1 is small and hardly readable in a standard article size. Kindly consider making the text larger, even if the Figure also will be larger.

2. Figure 4. It is not immediately clear from Figure 4, what the "score" and the "position" mean. Kindly add some explanation so the Figure is easier to understand. 

3. Figure 5. The same comments as for Figure 4.

Minor comments.

There are many spelling issues. Kindly consider using software to identify and fix them, e.g. Grammarly.com

1. Kindly fix the spelling for SARS-CoV-2 in the entire manuscript (line 19 etc).

2. Line 34. Kindly further comment on what is ORFs (-3).

3. LIne 60. Kindly check "peolple"

4. Line 69. Kindly consider using "and" instead of "&". Also lines 85, 102, 154, 158, 160, 

5. Line 72. Kindly check, whether it is "corona virus" or "coronavirus". Also lines 110-111, 245, 262, 280, 305, 309

6. Line 117. Kindly correct the number "1,36,566". Is it 136566?

7. Line 230. Kindly fix "Table: 6:"

8. Line 236. Is it "non epitopic" or "non-epitopic". Kindly check. Also, line 243

9. Line 244. Kindly check "Y449, , Q498, "

10. Lines 264-265. "co expression" is "coexpression, co-expression"

11. Line 333. Kindly check the "spike protein can able to elicit" part.

12. It is confusing that the authors claim no funding in the "Funding" section, and then acknowledge the government for Funding in the "acknowledgment" section. 

Reviewer 2 Report

In the manuscript "In silico screening of prospective MHC class I and II restricted T-cell based epitopes of 2 the spike protein of SARS-CoV-2 for designing of a peptide vaccine for COVID-19", the authors focused their attention on the use of vaccines administered against SARS-CoV-2 highlighting an alteration of the platelet count. In order to design an effective vaccine with high specificity and reduced side effects, the authors analyzed the antigenic epitopes of the spike protein of SARS CoV-2 involved in the initial binding of the virus with the angiotensin-converting enzyme-2 receptor (ACE-2 ) on the respiratory epithelial cells. They identified any epitopes suitable for the synthesis of an efficient peptide-based vaccine against the deadly COVID-19.

The work is very interesting and innovative. The results are explained very well, and the materials and methods are described in detail.

I would like to suggest the articles written by Zannella et.al (Design of Three Residues Peptides against SARS-CoV-2 Infection DOI:https://doi.org/10.3390/v14102103 ) and Galdiero et.al (SARS-CoV-2 vaccine development: where are we? DOI: 10.26355/eurrev_202103_25439) to improve the quality of the manuscript. It is also advisable to review the bibliography, for example in line 34 change [-3] with [1-3].
